# Understanding the connection between hospital goals and patient and family engagement: A scoping review

Umair Majid[1]*, Carolyn Steele Gray[1,2], Marianne Saragosa[2], Pia Kontos[3,4], Kerry Kuluski[1,5]

1 Institute of Health Policy, Management, and Evaluation, University of Toronto, Toronto, Ontario, Canada, 2 Lunenfeld-Tanenbaum Research Institute, Sinai Health, Toronto, Ontario, Canada, 3 KITE Research Institute, Toronto Rehabilitation Institute, University Health Network, Toronto, Ontario, Canada, 4 Dalla Lana School of Public Health, University of Toronto, Toronto, Ontario, Canada, 5 Institute for Better Health, Trillium Health Partners, Mississauga, Ontario, Canada

* umair.majid@mail.utoronto.ca

**Data Availability Statement:** All relevant data are within the paper and its Supporting information files.

## Abstract

The person-centered care movement has influenced hospitals to make patient and family engagement (PE) an explicit commitment in their strategic plans. This is often reflected in mission, vision, and value (MVV) statements, which are organizational artifacts intended to influence the attitudes, beliefs, and actions of hospital teams and employees because of their saliency in organizational documents and communications. Previous research has found that organizational goals for PE, like those articulated in MVV statements, can lead to effective and meaningful PE. However, a deeper understanding of how and under which circumstances MVV statements encourage and promote PE practices is needed. A scoping review was conducted to understand the connection between hospital PE goals (such as MVV statements) and PE processes and practices. The research question was: what is known about how hospital MVV statements relate to PE processes and activities? Following Arksey and O'Malley's scoping review approach, 27 articles were identified as relevant to the research question. These articles revealed five strategies that help realize hospital PE goals: communicating organizational goals; aligning documents that convey organizational goals; aligning organizational processes to support PE; providing employees with resources and support; and motivating and empowering employees to integrate PE into their work. We discuss the implications of misalignment between hospital goals and practices, which reduce team and individual motivation toward hospital PE goals.

## Background

Historically, healthcare organizations have paid little attention to or concerned themselves with mission, vision, and value (MVV) statements [1]. However, adopting MVV statements is vital for communicating strategic plans [2] and demonstrating an organizational commitment to patient and family engagement (PE) [3–6]. MVV statements represent the identity and culture of an organization [7, 8]. As organizational artifacts, MVV statements may influence the

**Funding:** This study was funded by the Canada Graduate Scholarship from the Canadian Institutes for Health Research. The funders had no role in study design, data collection and analysis, decision to publish, or preparation of the manuscript.

**Competing interests:** The authors have declared that no competing interests exist.

attitudes, beliefs, and actions of hospital teams and employees because of their saliency in organizational documents and communications [9, 10]. A vision statement outlines the future or desired state a hospital wants to reach; a mission statement indicates the hospital's current identity; and value statements represent the principles that guide health care activities [11, 12]. Hospitals may also present or communicate their goals in other ways, such as strategic plans, policy statements, a declaration of values, or a bill of rights and responsibilities.

The person-centered care movement has influenced hospitals to make PE an explicit commitment in their strategic plans. PE refers to the "ways in which patients can draw their experience and. . .apply their priorities to the evaluation, development, and organization and delivery of health services" [13]. Previous research has found that organizational goals for PE (such as those articulated in MVV statements) can lead to effective and meaningful PE [14–16]. For example, two studies found that MVV statements informed the development of a broad partnership framework or a code of conduct, which clarified employees' roles in engagement activities and understanding of the importance of PE to the organization [15, 16]. In this way, MVV statements can influence teams and individuals by enabling them to understand PE initiatives in relation to organizational goals. In this scoping review, we refer to PE goals as hospital PE goals, not program, team, or individual ones.

While a promising strategy, developing MVV statements alone is potentially insufficient for promoting and implementing PE in organizational activities, processes, and practices. Jick has noted that a vision statement is responsible for 10% of organizational change, and implementation is the rest [17]. In health care, hospitals are expected to communicate their commitment to PE and provide supporting evidence that they do so [18–20], including in hospital accreditation [21]. Clarifying the connection between hospital PE goals (e.g., having an explicit vision statement focusing on patient experience) and PE practices (e.g., engaging patients in hospital governance) will deepen our understanding of how and under which circumstances MVV statements encourage and promote PE practices and activities. To this end, a scoping review of research articles discussing PE interventions, strategies, tools, or outcomes can be insightful. Exploring the factors that influence the implementation and operationalization of hospital PE goals in PE practices can support health service planning and delivery to be effective and person-centered.

## Methods

### Approach

This scoping review follows Arksey and O'Malley's six-step approach: scoping; searching; screening; data extraction; data analysis; and an optional stakeholder consultation [22]. The scoping review extension of the PRISMA checklist was used to structure this scoping review [23].

### Scoping: Identifying the research question

Before conducting the systematic database search, the lead author (UM), in consultation with the research team, performed an exploratory search in MEDLINE to become familiar with the MVV literature and associated PE processes and activities. The search consisted of subject headings and keywords: patient participation/; patient engagement.mp.; Hospital Planning/; Quality Improvement/; vision.mp.; and mission.mp. Articles that discussed the connection between MVV statements and PE processes and activities were identified and used to refine the article eligibility criteria. This exploratory search informed a more comprehensive search strategy reviewed by an information scientist using the peer-review of electronic search strategies (PRESS) tool [24].

### Searching: Identifying relevant studies

A database search was conducted in MEDLINE, PsychINFO, Embase, and CINAHL using the search strategy available in S1 File on February 10, 2022. A search update was conducted on March 20, 2023. In consultation with the research team, UM performed a targeted grey literature search of corporate websites and journals about PE. The complete list of websites and journals searched is available in S1 File.

### Screening: Study selection

Eligible citations comprised MVV statements (or other ways to represent and communicate organizational goals) and the connection between MVV (or related concepts) statements and PE practices. These articles could have focused on concepts related to MVV statements and mentioned them in the title or abstract. See S1 File for a fuller account of the eligibility criteria for each screening step.

Two reviewers (UM, MS) pilot screened 100 articles with 90% agreement at the title and abstract screening stage. UM then reviewed all titles, abstracts, and full texts independently using the eligibility criteria available in S1 File with 15% verification from a second reviewer (MS). While complete verification by the second reviewer is the gold standard for review screening, we found over 95% agreement in the articles reviewed by two reviewers (UM, MS), which was an acceptable number [25]. This screening process is consistent with current methodological discussion suggesting that single screening can be effective when completed by an experienced reviewer familiar with the research topic [26]. All screening was performed through Covidence [27], and screeners resolved conflicts through discussion.

### Data extraction of study characteristics: Charting the data

In consultation with the research team, two reviewers (UM, MS) developed and tested a data extraction form to document the study characteristics of included articles (i.e., author, year of publication, country, title, research objectives, study design, data collection methods, number and type of participants, demographic characteristics of participants, and the overall findings). This information was used to understand the context and range of included studies. This data extraction form was pilot tested on three articles to capture various characteristics and contextual factors. Through discussion, UM and MS refined the data extraction form to achieve consensus. UM then extracted data from included articles independently with feedback from the research team.

### Data analysis: Collating, summarizing, and reporting the results

**Analytical framework.** Kantabtura's five realization factors served as the analytic framework for this study [28]. The framework originates from the service and leadership industry and describes the connection between MVV statements and organizational performance and practices. Each factor represents a category of strategies that enable and align organizational goals with practices. This framework was adopted to elaborate on strategies and practices needed to implement and operationalize hospital PE goals in PE practices. The five realization factors are shown in Table 1.

**Analysis procedures.** Deductive and inductive thematic analysis was used to identify and categorize findings from the included articles across two analytic stages [29]. The realization factors framework was used in the first stage to extract themes from the included articles [28]. Two reviewers (UM, MS) reviewed five of these articles to get a broad sense of data and to create a preliminary coding scheme that captured salient codes and concepts across the realization

**Table 1.**

| Realization Factor | Description |
|---|---|
| Communication | Ensuring that the right messages reach the right people at the appropriate times and places |
| Organizational Alignment | Reorganizing and tailoring governance structures, decision-making processes, policy frameworks, and resource allocation protocols to support MVV statements |
| Staff Personal Factor | Providing staff with resources and support to align their work with MVV statements |
| Motivation | Employee buy-in into MVV statements |
| Empowerment | Employee long-term commitment to MVV statements and goals |

factors as the themes. UM created a single data analysis template to aid in capturing codes and concepts from included articles. The analytic framework [28] provided the themes for this template and the preliminary coding scheme. In the second stage, inductive analysis was conducted to identify specific organizational strategies and outcomes for each realization factor. This approach was advantageous because the framework does not operationalize the realization factors into specific organization strategies and outcomes; to our knowledge, this framework has yet to be applied to the PE literature. All themes and subthemes were reviewed to develop narrative summaries that provided an interpretation of the connection between hospital PE goals and practices. UM consolidated all summaries, and the research team revised them accordingly.

**Quality appraisal.**   Quality appraisal of included studies was not conducted because scoping reviews do not require it [30].

## Stakeholder consultation: Optional consultation exercise

The research team includes experienced researchers who have conducted and published several research studies on PE and related topics [31–34]. Therefore, no stakeholder consultations were conducted.

# Findings

## Search results

The initial and updated searches combined yielded a total of 8992 abstracts after removing duplicates. Based on the eligibility criteria at the title and abstract screening stage, 8841 records were excluded. The remaining 151 articles from the database search and 42 from grey literature sources were screened at the full-text stage (total of 193 articles). The review team excluded 166 articles for different reasons shown in Fig 1. This review included 27 articles representing 26 studies [20, 35–60].

## Article characteristics and setting

All articles were published after 2011, with 12 (44.4%) published between 2011–2016 and 15 (55.6%) between 2017–2022. More than half of the included articles were published in the United States (n = 14, 51.9%) [20, 36, 37, 39, 40, 42, 44–48, 51, 52, 57] and adopted a qualitative study design (n = 15, 55.6%) [20, 35, 37–39, 41, 44, 49, 52, 54–59]. Other types of articles included descriptive reports (n = 5, 18.5%) [42, 45, 46, 50, 51], evidence syntheses (n = 2, 7.4%) [43, 47], quantitative study designs (n = 2, 7.4%) [48, 53], commentaries (n = 2, 7.4%) [36, 40], and mixed methods study (n = 1, 3.7%) [60]. Articles included a total of 443 participants, which consisted of 207 (46.7%) administrators, directors, and managers across eight studies

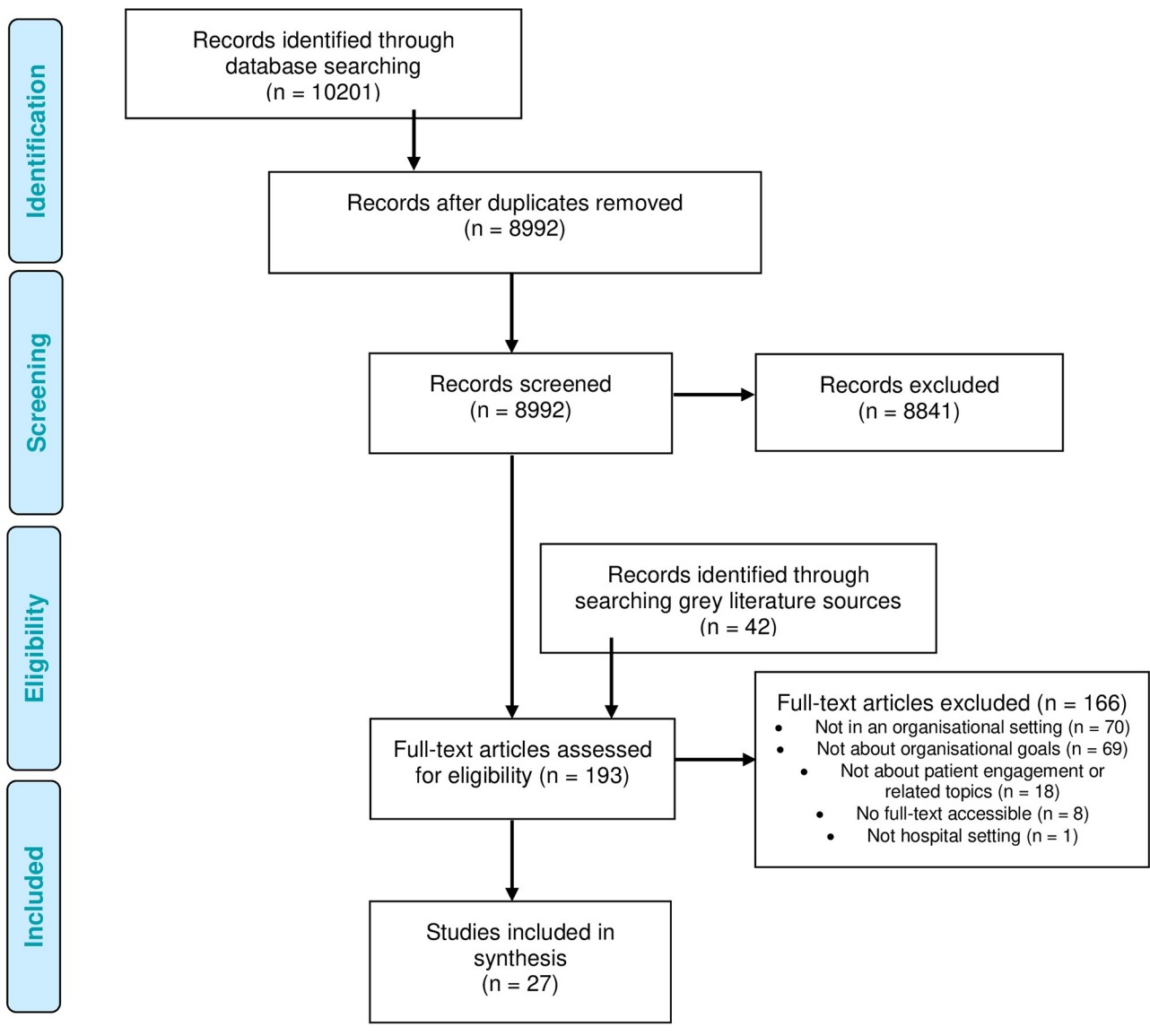

**Fig 1. Screening and selection process.**

[37–39, 49, 53, 55, 58, 60]; 55 (12.4%) leaders across three studies [37, 38, 44]; 54 (12.2%) patient, family or community advisors across four studies [38, 55, 57, 59]; and 82 (18.5%) clinicians across five studies [37, 38, 49, 58, 59]. More articles named a specific hospital, hospital system, program, or initiative (n = 16, 59.3%) [35, 38–43, 45–47, 50, 51, 54, 56, 58, 59] than articles that described multiple hospitals or hospital systems, or discussed hospitals generally (n = 11, 40.7%) [20, 36, 37, 44, 48, 49, 52, 53, 55, 57, 60]. The complete list of study characteristics for each included article is shown in the SI File.

## Synthesis

This section describes the synthesis of findings aligned with the realization factors [28]: communicating hospital goals, aligning goals, aligning processes, employee personal factors; and motivation and empowerment. Table 2 provides a summary of the barriers and strategies.

**Table 2. Summary of barriers and strategies.**

| New Realization Factors Based on Review Findings | Barrier and Strategy | Description |
|---|---|---|
| **Communication**: Ensuring that the right message reaches the right people at the appropriate times and places | None | Effective leadership communication, utilizing both verbal and non-verbal strategies, is crucial for employees to understand and implement PE goals. |
| **Aligning Goals**: Aligning goals across documents, strategies, and behaviors | None | Goal alignment across documents, strategies, and employees facilitates PE interventions, but variability and tension among competing goals and leadership perspectives can hinder implementation. |
| **Aligning Processes**: Reorganizing and tailoring organizational structures (e.g., governance, decision-making process, and resource allocation protocols) to support the realization of PE goals | Barrier: External Pressures | External pressures such as regulations and financial constraints can impede the hospital's alignment with PE goals, although patient-centered legislation can facilitate this alignment. |
| | Strategy 1: Intra- and Inter-Organizational Collaboration | Intra- and inter-organizational collaboration, through synergies and team-building, generally supports PE implementation, although some leaders question its necessity for structural reorganization. |
| | Strategy 2: Committees | Committees, particularly Patient and Family Advisory Councils (PFACs), serve as a strategy to align structures and processes with PE goals, and their inclusion of patients, families, and community members enhances buy-in and effective implementation. |
| | Strategy 3: Hiring | Strategic hiring practices, including the creation of PE-specific positions and the recruitment of individuals with PE competencies, along with the involvement of patients and families in the hiring process, contribute to aligning the workforce with hospital PE goals and values. |
| | Strategy 4: Employee Performance | Linking employee performance objectives to PE goals and involving PFACs in setting behavioral standards serve as strategies to align employee performance with hospital PE objectives. |
| | Strategy 5: Restructuring Workflow and Practices | Opinions on restructuring workflow and practices as a strategy for PE implementation are divided; while some leaders deem it unnecessary, other studies advocate for its importance. |
| **Employee Personal Factor**: Supporting hospital employees to integrate PE into their practices by providing them with resources, tools, support, space, and incentives | Strategy 1: Financial Resources | Financial resources impact PE strategy, with lack of employee engagement attributed to financial disincentives and fee-for-service funding schemes posing challenges to the integration of PE. |
| | Strategy 2: Training | Training programs, ranging from introductory sessions to specialized communication training, serve as a comprehensive strategy to align employee behaviors with PE goals, although the specifics of professional development programs remain unclear. |
| Employee beliefs and the relevance and meaning of goals to their professional practices | Strategy 3: Tools | Implementation support through PE readiness assessments and tools serves as a targeted strategy to facilitate PE goals. |
| | Strategy 4: Physical Space and Technologies | Improvements in physical space, including design elements and technological infrastructure, serve as a multifaceted strategy to support PE goals by enhancing both the aesthetic and functional aspects of the hospital environment. |
| **Motivation**: Strategies used initially to obtain buy-in and commitment to hospital PE goals | Barrier: Resistance to Change | Resistance to change as a barrier is fueled by factors such as competing priorities, technical focus, and employee feelings of being unheard, but can be mitigated through patient and family involvement and a supportive environment that fosters alignment with PE goals. |

(*Continued*)

**Table 2.** (Continued)

| New Realization Factors Based on Review Findings | Barrier and Strategy | Description |
|---|---|---|
| **Empowerment**: Strategies to secure a long-term commitment to PE processes and activities | Strategy 1: The Role of Leadership | The role of leadership is instrumental in setting the tone for PE, as leaders who explicitly and implicitly support PE goals motivate employee engagement and can influence hospital-wide practices through various behaviors, including active participation in committees and effective communication. |
| | Strategy 2: Employee Motivation Strategies | Employee motivation strategies include alignment with hospital PE goals, focusing on small wins for buy-in, involving patient advisors and PFACs for ongoing integration, and developing vision statements through employee consultation to foster ownership of PE objectives. |
| | Strategy 3: Employee Empowerment Strategies | Employee empowerment strategies focus on aligning care with PE goals to prevent role strain and burnout, using recognition to promote commitment, and employing data collection and progress reporting as tools for empowerment. |

**Operationalization of the realization factors.** The original realization factors framework lacks clear operationalization because it only offers broad definitions of each factor and a few examples of organizational strategies. In the following sections, based on our scoping review, we detail specific organizational strategies and outcomes under each realization factor to help unpack the connection between hospital goals (MVV statements, strategic plans, and other representations of goals) and PE practices (e.g., aligning goals across organizational documents, elaborating on strategies that align processes with hospital PE goals, and describing ways to provide support and resources to employees to integrate PE into their work). This approach entailed operationalizing the realization factors in terms of strategies and activities that reflect the work of teams and employees, strengthening our understanding of how the realization factors relate to or support PE practices.

Not all included articles were explicit about the connection between hospital goals and PE processes and activities. While some articles discussed how explicit hospital PE goals (e.g., having an explicit vision statement focusing on patient experience) influenced PE practices (e.g., engaging patients in hospital governance), other articles discussed how hospital goals with some PE elements (e.g., indirect statements about collaborating with or serving communities) influenced PE practices. The following sections present our analysis of both instances to unpack the connection between hospital goals and PE practices and processes. We have specified detailed findings about the relationship between PE goals and practices wherever possible. Furthermore, staying consistent with the authors' descriptions of organizational goals was not easy, which differed across articles. For example, articles discussed MVV statements [45, 47, 49, 50], strategic priorities [37, 45, 50, 52, 53], values and care principles [38, 53], and a "community orientation" [48]. However, incorporating different representations of organizational goals allowed us to create a broader representation of organizational strategies. Therefore, in this scoping review, PE goals refer to different representations of hospital PE goals, not program, team, or individual ones. It is important to note that we found considerable overlap between realization factors because they were discussed in the same, similar, or related ways. Table 3 provides a summary of the findings and an overview of how the realization factors were modified based on the findings of this review.

**Communicating hospital goals.** Communication, as defined by the realization factors framework, ensures that the messages reach employees and other stakeholders at the appropriate times and places [28]. Communicating hospital goals helps employees understand how their daily routines and professional practices may change to support PE [36, 58]. Seven

**Table 3. Summary of the findings across the realization factors framework.**

| Original Realization Factors from Analytic Framework | New Realization Factors Based on Review Findings | Themes | Findings |
|---|---|---|---|
| Communication | **Communication**: Ensuring that the right message reaches the right people at the appropriate times and places | | • Communicating PE goals enables employees to understand the meaning and how their daily routines will change [36, 58] <br> • The leadership was responsible for communicating goals effectively [42, 46, 47, 50, 55, 58] <br> • Verbal and non-verbal strategies were useful for communicating PE goals [37, 41–43]. |
| | **Aligning Goals**: Aligning goals across documents, strategies, and behaviors | | • Goal alignment between documents, strategies, and employees was a facilitator of PE interventions [41, 43, 45, 49, 52]. <br> • Varying goals stymied goal alignment across [43, 53, 56]. <br> • Goal alignment was important for implementing PE [42, 44, 45, 46, 52, 54, 56]. <br> • Employees look toward goals and strategies to guide their daily work; they inform employee behaviors, practices, and attitudes [35, 36, 38]. <br> • Tension or confusion between competing hospital goals adversely affected employee behaviors [38, 44, 47, 59, 60]. <br> • Variability between leaders on the importance of PE [60]. <br> • Employees show agency when addressing incongruence between hospital goals and their practices [35]. <br> • While hospital PE goals were common, implementation of PE was lacking [53, 57]. |
| Organizational Alignment | **Aligning Processes**: Reorganizing and tailoring organizational structures (e.g., governance, decision-making process, and resource allocation protocols) to support the realization of PE goals | Barrier: External Pressures | • Regulations and financial pressures restricted the hospital's ability to align structures and processes with PE goals [37, 38, 44, 49, 52, 55, 60]. <br> • Regulatory goals and government agendas were contrary to patient-centered care [37]. <br> • Patient-centered legislation, targets, and policies facilitated aligning structures and processes with PE goals [55]. <br> • Financial pressures that were a barrier to aligning structures and processes to PE goals included productivity, competitive external environment, and internal budget constraints [44, 46, 52]. |
| | | Strategy 1: Intra- and Inter-Organizational Collaboration | • Establishing synergies between existing services and initiatives, integrating PE in hospital-wide initiatives, and building collaborative teams to support PE implementation [37, 40, 49]. <br> • Leaders in one study believed that collaboration was not necessary for reorganizing structures and processes for PE [44]. |
| | | Strategy 2: Committees | • PFACs were a strategy to align structures and processes to support hospital PE goals [42, 43, 45, 51]. <br> • A survey study found an association between PE in decision-making activities and the implementation of PE across the hospital [53]. <br> • Important to have patients, families and community members in existing committees and decision-making processes; this increased buy-in, communication, and adoption of hospital goals [38, 42, 43, 47, 50, 52]. |
| | | Strategy 3: Hiring | <u>PE-Specific Positions</u> <br> • Patient navigators, Chief Experience Officer, and customized vase managers [40, 43, 49]. <br> <u>Hiring People with PE Competencies and Values</u> <br> • A new CEO with PE values led to the hiring of executives and employees that shared the PE values [47]. <br> • Revising hiring policies to focus on customer-oriented care [39]. <br> • Changing position descriptions and requiring applicants to complete a behavior assessment to identify PE values [39, 42]. <br> <u>Identifying and Promoting Existing Employees</u> <br> • Important to identify existing employees with PE values [36, 37, 50, 52]. <br> • Challenge in maintaining stable teams [36, 49]. <br> <u>Patient and Family Involvement in Hiring</u> <br> • Conveys a solid commitment to PE goals [36]. <br> • Allows hiring of people with PE values [38]. |
| | | Strategy 4: Employee Performance | • Tying employee performance objectives and behavioral standards to hospital PE goals [36–38, 42]. <br> • Involving PFACs in developing employee behavior standards [36, 38]. |
| | | Strategy 5: Restructuring Workflow and Practices | • Leaders believed that restructuring workflow and practices was not necessary [44]. <br> • Other studies emphasized restructuring workflow for implementing PE [48, 52]. |

*(Continued)*

**Table 3.** (Continued)

| Original Realization Factors from Analytic Framework | New Realization Factors Based on Review Findings | Themes | Findings |
|---|---|---|---|
| **Staff Personal Factor** | **Employee Personal Factor**: Supporting hospital employees to integrate PE into their practices by providing them with resources, tools, support, space, and incentives<br><br>Employee beliefs and the relevance and meaning of goals to their professional practices | Strategy 1: Financial Resources | • Lack of employee engagement in PE goals was attributed to financial disincentives [49].<br>• Fee-for-service funding scheme opposed the integration of PE in practice [47, 49, 52]. |
| | | Strategy 2: Training | • More than three-quarters of European hospitals surveyed by one study provided training to hospital employees [53].<br>• Training programs enabled employees to align their behaviors and practices with hospital PE goals [36, 37, 46].<br>• *Introductory sessions* and *new employee orientation sessions* introducing PE and related concepts to employees were an essential first step in the training process [37, 44].<br>• Communication training programs focused on addressing patient and family complaints, offering comfort measures to families as they arrive, anticipating patient and family needs, and listening for values and desires [37, 39, 43].<br>• Professional development and support programs were important, but no detail was provided on the nature of these programs [36, 43, 44, 49]. |
| | | Strategy 3: Tools | • Providing implementation support through PE readiness assessments and tools [45]. |
| | | Strategy 4: Physical Space and Technologies | • Several strategies were mentioned by studies to improve the physical design of hospitals, such as increasing natural lighting, avoiding clutter in the hallways and behind nursing units, including clear signage for navigating in the hospital, and having visually appealing decorations [39, 41, 43, 49].<br>• The physical space was enhanced by the support of a technological infrastructure such as medical equipment, well-coded electronic medical record, information technology systems, and data resources for recording progress [47, 49, 51, 52]. |
| **Motivation** | **Motivation**: Strategies used initially to obtain buy-in and commitment to hospital PE goals<br><br>**Empowerment**: Strategies to secure a long-term commitment to PE processes and activities | Barrier: Resistance to Change | • Factors that encouraged employee resistance to change included:<br> • Feeling unheard because patients and families were prioritized in hospital goals and activities [38].<br> • Physician's ambivalent attitude toward PE [52].<br> • Uncertainty or ambiguity on how PE may change practices [52].<br> • Competing hospital priorities [55, 59, 60].<br> • An emphasis on technical aspects of care [41].<br> • An overfocus on achieving small wins caused a loss in focus on more extensive cultural transformation [41].<br>• Working with patients and families on PE initiatives and PE goals reduced employee resistance to change [37, 38, 41, 50].<br>• Creating a supportive environment was critical to changing employee attitudes towards PE goals and implementing PE practices across hospital functions [37, 38, 41, 45, 50]. |
| **Empowerment** | | Strategy 1: The Role of Leadership | • Leaders who expressed implicit and explicit support for PE goals were a strong motivator for employees to engage in PE [37–39, 42, 43, 46, 49, 52, 55, 58, 60].<br>• Leadership expressed support for PE goals in different ways. Certain leadership behaviors set the tone for PE across the hospital [1, 4–6, 8–11, 15, 19, 23, 36, 37, 40, 41, 43–45, 46, 47, 50, 60].<br>• Leaders could set the tone by modelling PE in their activities, such as through active participation in PE committees [37], encouraging policies and procedures that encourage PE [36, 44, 46, 58], communicating about PE initiatives and how each employee contributes to them [46], communicating PE goals [47], showing the benefits of patient partnerships for each employee [50], and being aware of challenges that employees will face in implementing PE practices [44, 50]. |
| | | Strategy 2: Employee Motivation Strategies | • Employees who identified with hospital PE goals were more likely to be motivated toward integrating PE into their practices [38].<br>• Focusing on achieving small wins was an effective strategy for obtaining widespread employee buy-in [38, 49].<br>• Patient advisors and PFACs who can support the ongoing integration of PE in practices and share their stories and narratives were used as an initial strategy to promote PE goals [38, 55, 57].<br>• Developing vision statements through consultation with employees helped promote ownership of PE goals across hospital functions [41]. |
| | | Strategy 3: Employee Empowerment Strategies | • Providing care not aligned with PE goals led to role strain and burnout over time [56].<br>• Recognition promotes employee commitment to PE goals [40–43].<br>• Collecting data and reporting progress empowered employees [38, 40, 43]. |

articles highlighted the connection between goal communication and PE practices [36, 42, 46, 47, 50, 55, 58]. Specifically, leaders played a critical role in goal communication by clarifying how each employee and team contributed to hospital goals [46, 58] and how they may benefit from implementing PE in their work [50]. Goals were communicated through verbal and non-verbal strategies for communicating PE goals, such as displaying goals in the reception area, using computer screen savers and hospital websites, and using email signatures [37, 41–43]. However, none of the articles examined whether communicating hospital goals made a difference in implementing PE practices.

**Aligning goals.** Eighteen articles discussed aligning goals between documents, strategies, and employees [35, 36, 38, 41–47, 49, 52–54, 56, 57, 59, 60]. Aligning goals this way was described as a facilitator of PE interventions by influencing employee attitudes and behaviors [41–46, 49, 52, 54, 56]. Studies mentioned how goals and strategies could potentially guide employees' daily work and inform their behaviors, practices, and attitudes [35, 36, 38]. However, different hospital goals (e.g., financial productivity, patient experience optimization, technological innovation, health care providers protection, and community development) weakened the alignment between organizational strategies and employees [43, 52, 56, 59, 60]. According to three articles, employees were less likely to adopt attitudes and behaviors related to PE if multiple goals were communicated that employees perceived to be different messages (i.e., financial productivity versus patient experience) [38, 44, 47, 59] or if there was variability among leadership on the importance of PE [60]. Hospitals also faced the challenge of aligning their PE mission statement with the thousands of employees' diverse roles, practices, and values [47]. While one article described alignment between goals, strategies, and employees led by hospital leadership [47], another article emphasized the agency and autonomy of employees in addressing incongruence between PE goals and practices [35]. In this example, providers and staff perceived incongruence between their hospital's goal of improving patient experience and the standardized clinical care they provided, which limited their ability to enhance the patient experience [31]. In response to this incongruence, staff, and providers worked harder at the clinical interface to improve care quality and patient experience [35].

One striking example of the incongruence of goals is demonstrated in one article examining the differences between three hospital documents [56]. The first type of document is referred to as higher-order documents and conveys organizational goals (e.g., mission and vision statements). The second type is mid-range documents, which identify how goals in higher-order documents are implemented in practice (e.g., strategies, policies, procedures, and organizational models). The third type, care documents, guided the interactions between employees, patients, and family (e.g., care plans, assessment charts, and progress notes). This article found incongruence between higher-order and care documents, indicating a misalignment between PE goals and practices [56]. This misalignment was apparent in the language of each document type; care documents emphasized care standardization, which contradicted the PE goals conveyed in higher-order documents [56]. Such misalignment limited employee understanding of hospital PE goals and their ability to ensure that their work is consistent with the goals. The authors suggested aligning the hospital's "documented systems of care" to minimize the tension and incongruence between goals and practices [56]. However, it is unclear whether having hospital PE goals promotes PE practices and interventions. One study could not conclude whether hospitals with hospital PE goals were more likely to implement PE practices than hospitals with only financial goals and without a hospital commitment to PE [57].

In summary, aligning goals between documents, strategies, and employees facilitated PE. However, the evidence in this review suggests that aligning goals is rarely achieved because of various factors, such as diverse goal messages. Aligning goal messages across organizational

documents can promote employee understanding of PE and how they can integrate it into their work.

**Aligning processes to hospital goals.**   From our inductive analysis, this section elaborates on external forces, intra- and inter-organizational collaboration, committees, hiring, employee performance assessment, and restructuring workflow and practices. Nineteen articles referred to aligning processes [20, 33–38, 40–43, 45–49, 51, 53, 55], which refers to how hospitals reorganize and tailor organizational structures to support PE goals according to the realization factors framework [27]. Included articles described one barrier (external forces referring to restricting alignment between structures and processes with PE goals) and five strategies to align processes with goals (intra- and inter-organizational collaboration, committees, hiring, employee performance assessment, and restructuring workflow and practices).

*Barrier*: *External forces*. Seven articles mentioned two external forces that restricted the ability of hospitals to align structures and processes with PE goals: financial pressures; and regulatory policies [37, 38, 44, 49, 52, 55, 60]. Three articles described productivity pressures and budget constraints as barriers to aligning structures and processes with PE goals [44, 52, 55]. Regulatory policies (i.e., regulations, rules, plans, and benchmarks) and government agendas contradicted patient-centered care because they emphasized care standardization [37, 38, 49, 52]. On the other hand, one article explained how having patient-centered legislation, targets, and policies facilitated alignment between hospital structures, processes, and PE goals [55]. However, alignment was only possible in this situation if hospitals invested in infrastructure that sustained PE [55].

*Strategy 1*: *Intra- and inter-organizational collaboration*. Six articles discussed collaboration within and outside the hospital as a strategy to align structures and processes with PE goals [37, 40, 43, 44, 49, 52]. Three articles mentioned synergizing existing services and initiatives, integrating PE in hospital-wide initiatives, and building collaborative teams with other health service organizations to support PE [37, 40, 49]. However, leaders in one article believed that collaboration was unnecessary in process alignment [44]. In contrast, several authors reported challenges in implementing PE, which may have been attributed to the separation of hospitals from other organizations [37, 44]. For example, one study indicated that having an integrated care infrastructure (e.g., partnerships with primary care practices, involving community health groups, and using telehealth to access and provide care) promoted PE practices [43].

*Strategy 2*: *Committees*. Four articles established Patient and Family Advisory Committees to align hospital structures and processes to support hospital PE goals [42, 43, 45, 51]. Hospital administration created Patient and Family Advisory Committees to support and encourage PE initiatives across hospital functions. They did this by differentiating between PE activities, developing organizational models for PE, reviewing evidence-based PE practices, and making recommendations for hospital design, structure, process, and implementation [43, 45, 51].

Eight articles emphasized the importance of involving patients, families, and community members in decision-making committees to support PE goals [38, 42, 43, 47, 50, 52, 53, 57]. One study reported an association between PE in hospital governance committees and greater implementation of PE across hospital functions [53]. Interestingly, the tension between competing hospital priorities was alleviated by involving patients and families, which also increased employee buy-in to hospital PE goals [38, 57]. Specific strategies that achieved these goals included sharing the engagement process on hospital websites [38], listening and considering the perspectives of patients and care partners during committee meetings [42], and having multidisciplinary committees with a broad range of experiences and perspectives from patient and family partners [42].

*Strategy 3*: *Hiring*. Thirteen articles mentioned the importance of modifying hiring processes to support PE goals [36–40, 42, 43, 45, 47, 49, 50, 51, 52]. Three articles mentioned

creating PE-specific positions, such as roles dedicated to transforming the workplace culture [43], patient navigators [43], a Chief Experience Officer [40], and customized case managers to provide hospital units with tailored support for PE implementation [49]. On the other hand, six articles emphasized hiring people with PE competencies and values [36, 38, 39, 42, 47, 51]. One article explained how a new CEO with strong PE values led to hiring other employees and executives who shared similar values, which ultimately supported a hospital's PE vision [47]. Articles recommended revising hiring policies to focus on customer-oriented care [39], aligning position descriptions with PE values [39, 42], requiring applicants to complete a behavioral assessment that identifies applicants with PE values [39, 42], and having patient and family partners identify applicants who shared the hospital's PE goals and values [38]. Furthermore, four articles mentioned the importance of identifying and promoting existing employees with PE values to support hospital PE goals [36, 37, 49, 52]. However, articles mentioned that employee turnover might disrupt PE programs and practices [36, 49]. Articles that discussed hiring practices emphasized the importance of an organizational culture that supports and promotes PE.

*Strategy 4*: *Employee performance assessment*. Five articles discussed modifying employee performance assessment processes to support hospital PE goals [36–38, 40, 42]. Several articles mentioned tying employee performance objectives and behavioral standards to hospital PE goals [36–38, 42]. For example, a hospital system added "patient/customer focus" as a behavioral expectation for all employees [42]. Involving hospital Patient and Family Advisory Committees in developing employee behavior standards and using patient feedback during performance reviews strengthened the connection between employee performance and hospital PE goals [36, 38].

*Strategy 5*: *Restructuring workflow and practices*. Six articles generally mentioned restructuring workflow and employee practices [20, 35, 44, 45, 48, 52]. For example, one article described changing workflow to incorporate PE [52]. However, included studies discussed this topic generally and provided few details on strategies or approaches to restructuring workflow. Leaders in one article believed that restructuring workflow and practices were optional for PE [44]. However, only some leaders in the article expressed this belief, while others emphasized restructuring workflow for PE generally [44, 48, 52].

In summary, the included articles described the importance of aligning processes to support hospital PE goals and activities across one barrier (external forces) and five strategies (collaboration, committees, hiring, employee performance assessment, and restructuring workflow and practices).

**Employee personal factors.** Seventeen articles mentioned the employee personal factors [36–41, 43–50, 52, 53, 55], which includes employee beliefs and hospital resources that support employees to integrate PE in their work according to the realization factors framework [28]. Through inductive analysis, we elaborated on the employee personal factors by describing strategies and employee perceptions of hospital PE goals and the importance of resources in integrating PE across hospital functions.

Participants described their views on the importance of resources for integrating PE into their work. Resources (e.g., providers, support staff, and physical space) ensured that hospitals integrated patient partnerships systematically across hospital functions [47, 50]. Several articles cited insufficient staff as barriers to implementing PE practices [20, 48, 52]. They reported needing additional time for training, supporting, and helping employees understand the scope of their professional responsibilities [20, 46, 49, 50]. Leaders and the human resources department were essential for improving care quality and patient experience because they developed PE models and incentives for integrating PE into hospital practices [36, 48].

However, across the articles, there were differing perspectives on the importance of resources for implementing PE [44, 47]. Larger hospitals tended to have more staff and resources to devote time to achieving PE goals [48]. One article described how employees who desired more resources to accomplish PE goals reflect a clinical culture emphasizing technological solutions [47]. However, another study mentioned that providing additional resources may encourage the use of standard questions and scripts in clinical care, which may oppose PE goals [44]. This remaining section is divided into the following strategies: financial resources, training, tools, and physical space and technologies.

*Strategy 1*: *Financial resources*. Five US-based articles mentioned the importance of financial resources in supporting employees to implement PE in their work [20, 45, 47, 49, 52]. These articles attributed the lack of employee engagement in hospital PE goals to a lack of financial incentives [49]. Furthermore, the fee-for-service funding scheme was a barrier to PE goals because it encouraged care volume over care quality and patient experience [47, 49, 52]. Two articles also emphasized addressing employee turnover by providing financial incentives to employees for integrating PE into their work [20, 45].

*Strategy 2*: *Training*. Fourteen articles mentioned training employees to incorporate PE practices [36–41, 43, 44, 46, 47, 49, 50, 51, 53]. More than three-quarters of European hospitals provide training to hospital employees in PE [53]. The training enabled employees to align their behaviors and practices with hospital PE goals [36, 37, 46]. Regular, continuous, and tailored training reinforced PE practices [37, 38, 43, 44, 49]. One article reported that training developed employees' ability to notice when policies and practices were not aligned with hospital PE goals [38]. Employees developed these abilities through reflection during training, such as discussing the emotional and social aspects of health care, and the importance of hearing patient and family stories [37, 41]. Four types of training programs were mentioned in the articles: introductory programs; new employee orientation; communication training; and professional development and support programs (Table 4).

*Strategy 3*: *Tools*. Tools, such as a guide for engaging patients and families, were also crucial for implementing PE, but studies did not discuss tools in detail [45, 47, 51]. Conducting PE readiness assessments for hospital departments was an important step before integrating PE

**Table 4. Training programs.**

| Training | Description |
|---|---|
| Introductory and New Employee Orientation | • Introduce PE and related concepts to new and existing employees [36, 37, 43, 44, 47]<br>• Sharing patient stories and how PE enriches the work [36]<br>• Describing patient-centered values and principles [43]<br>• Co-leading introductory and new employee orientation training with patients and families [43] |
| Communication Training | • Addressing patient and family feedback, offering comfort measures to families as they arrive, anticipating needs, and listening to values and desires [37, 39, 43]<br>• One article focused on training related to a Patient Courtesy Initiative that provided employees with a sequence of steps for their interactions with patients: entering the room; making introductions; conveying information to patients and families; and responding to patient and family inquiries [39]. |
| Professional Development and Support Programs | • Enabled employees to align their professional goals with PE practices [36, 43, 44, 49, 51]<br>• No detail was provided on the nature of these programs |

systematically into their work [45]. In one hospital, readiness assessments supported the development of a customized charter for each department to guide alignment between hospital PE goals and department activities [45].

*Strategy 4*: *Physical space and technologies*. Four articles mentioned that the physical design of hospitals could encourage PE practices in a way that aligned with hospital PE goals [39, 41, 43, 49]. One article describes an onstage/offstage design of a new patient-centered hospital [39]. The onstage area included what patients saw: hallways, rooms, and units; the offstage area was blocked off for all patients and contained the technical aspects of care provision, such as medical equipment; and the concierge aspect referred to the responsive nature of customer service [39]. This hospital was designed to encourage employees to communicate with patients by reducing the space where employees communicated with each other [39]. Articles mentioned improving the physical design of hospitals by increasing natural lighting, avoiding clutter in hallways and behind nursing units, including clear signage and directions around the hospital, and having visually appealing decorations [39, 41, 43, 49]. The physical space was also enhanced by the support of technological infrastructure, including medical equipment, well-coded electronic medical records, information technology systems, and other data sources [47, 49, 51, 52]. These were described as strategies that encouraged PE practices in a way that aligned with hospital PE goals.

In summary, included articles discussed employee personal factors, such as employee perspectives towards hospital resources, training, tools, physical space, and technologies. However, few details were provided on how these strategies strengthen the connection between hospital PE goals and practices.

**Motivation and empowerment.**   Seventeen articles referred to motivation and empowerment [36–47, 49, 52, 55–57]. According to the realization factors framework, motivation refers to the strategies used to obtain buy-in to hospital PE goals, and empowerment refers to the strategies used to secure long-term commitment [28]. From our inductive analysis of the articles, this section discusses one barrier (resistance to change) and three strategies: the role of leadership, employee motivation strategies, and employee empowerment strategies.

*Barrier*: *Resistance to change*. Seven articles identified employee resistance to change as a critical barrier to integrating PE into hospital practices [38, 41, 49, 52, 55, 59, 60]. Factors that encouraged employee resistance included feeling that patients and families were prioritized over them in all hospital goals and activities [38], ambivalent attitudes toward PE [52], uncertainty or ambiguity on how PE will change practices [52], competing hospital priorities [55, 59, 60], emphasis on the technical aspects of care [41], and the emphasis on 'small wins' at the expense of organizational and cultural transformation [41].

Working with patients and families on PE initiatives or developing PE goals reduced resistance to change [37, 38, 41, 50], led to a common understanding of PE in the hospital [38], increased employees' value of experiential knowledge [39], developed employees' ability to identify disconnections between PE goals and care practices [41], created like-mindedness between employees and patients [50], and fostered willingness to try new things [52]. Leaders and team members who created a supportive environment that allowed employees to learn, develop, and incorporate PE in their work were critical to changing employee attitudes toward PE goals and implementing PE practices across hospital functions [37, 38, 41, 45, 55].

*Strategy 1*: *The role of leadership*. Seventeen articles mentioned the importance of leadership in motivating and empowering employees toward PE goals [36–47, 49, 52, 55, 58, 60]. Leaders who expressed implicit and explicit support for PE goals motivated employees to engage in PE [37–39, 42, 43, 47, 49, 52, 55, 58, 60]. Positive leadership behaviors seemed to model PE across the hospital [36, 37, 40, 41, 43–47, 50, 60], such as active participation in PE committees [37], encouraging policies and procedures that promoted PE [36, 44, 46, 58], communicating PE

initiatives and each employee's contributions to these [46], sharing PE goals [47], showing the benefits of patient partnerships for the hospital and professional practices [50], and being aware of the challenges employees face implementing PE in their work [44, 50]. Articles mentioned that highly persistent leadership that imbued inspiration and provided support could motivate and empower employees to integrate PE into their practice in a way that aligns with hospital PE goals [37, 43, 50, 55].

*Strategy 2*: *Employee motivation.* As defined by the realization factors framework, motivation refers to strategies for obtaining buy-in to hospital PE goals [28]. Seven articles mentioned strategies for employee buy-in of hospital PE goals [37, 38, 41, 44, 49, 55, 57]. Three articles mentioned the importance of aligning employee (clinicians, frontline providers, and administrators) practices and personal values with hospital PE goals [38, 44, 49] because employees who identified with hospital PE goals were more motivated to integrate PE into their practices [38]. Focusing on achieving small wins effectively obtained widespread employee buy-in [37, 49]. This involved breaking goals into smaller, more manageable goals and celebrating small achievements [49]. Patient advisors and Patient and Family Advisory Committees promoted hospital PE goals by sharing their stories and supporting the ongoing integration of PE in practices [38, 55, 57]. Finally, developing vision statements through employee engagement promoted ownership of hospital PE goals [41].

*Strategy 3*: *Employee empowerment.* As defined by the realization factors framework, empowerment refers to strategies to secure long-term commitment [28]. Seven articles mentioned strategies that can empower employees to integrate PE into their practice in a way that aligns with hospital PE goals [38, 40–43, 49, 56]. Employees felt role strain and burnout when required to provide care that was not aligned with hospital PE goals [49, 56]. Four articles described recognition strategies that promoted employee commitment to PE goals [40–43], such as recognizing individuals who have excelled in integrating PE practices into their work and events that acknowledged and celebrated progress (e.g., employee BBQ events, health and well-being programs, and monthly forums) [43]. The second strategy that empowered employees was collecting data and reporting progress [38, 40, 43]. The third strategy was reinforcing accountability for PE practices at all levels [40, 42]. One article described how healthcare providers not being held accountable for patient-centered care conveyed organizational ambivalence towards PE goals, leading to lower employee commitment to hospital PE goals [37].

In summary, strategies that motivate and empower employees are essential for strengthening the connection between hospital PE goals and practices. However, employees may express resistance to change that can be addressed by working with patients and families on hospital initiatives and providing them with the tools to support PE goals and leadership behaviors that promote and help set the tone for PE at the hospital.

## Discussion

### Summary of the findings

In this scoping review, we unpacked the connection between hospital PE goals and PE practices. Using the realization factors framework as a guide [28], we elaborated on specific organizational strategies and outcomes related to goal communication, goal alignment, process alignment, employee personal factors, and motivation and empowerment. In this way, we operationalized the realization factors to explore how they relate to or support PE practices by elaborating on organizational strategies and outcomes.

This study expands on the existing literature examining how employees understand and internalize hospital PE goals (such as MVV statements) in their practices [2, 61, 62]. To our knowledge, this is the first evidence synthesis exploring the connection between MVV

statements and PE practices, which has provided a deeper understanding of MVV statements in the context of person-centered care, patient experience, and PE. Hospitals are encouraged to communicate their commitment to PE and provide evidence of PE activities [18–20], including in hospital accreditation [21]. This scoping review has contributed to understanding how and under which circumstances hospitals with PE goals influence or promote PE practices. However, the connection between hospital PE goals and practice was not always explicitly stated in included studies. Included articles were more explicit in the connection between communicating hospital PE goals, aligning goals across documents, motivating and empowering employees, and PE practices. However, no research found an association between these realization factors and PE practices. On the other hand, studies were implicit about the relationship between aligning processes and PE practices. There is a need for further research that specifically focuses on exploring the connection between hospital PE goals and PE practices.

## What are promising practices for promoting hospital PE goals and practices?

Previous research has shown that developing and communicating PE goals is essential for promoting PE processes and activities. For example, the Measuring Organizational Readiness for Patient Engagement (MORE) tool includes three items on developing and sharing an organizational vision for PE with employees, patients, and the public [63]. Similar to the MORE tool, studies in this scoping review point to the importance of hospital goals in promoting PE processes and activities and communicating the importance of hospital PE goals for integrating PE across units and functions to employees, patients, families, and other groups [36]. Goals were communicated through verbal and non-verbal strategies, such as displaying goals in the reception area, using computer screen savers and hospital websites, and using email signatures [37, 41–43]. However, none of the articles examined whether communicating hospital goals made a difference in implementing PE practices.

The findings of this scoping review align with the Engagement Capable Environments Framework (ECE), which consists of three pillars of meaningful PE in organizations: enlisting and preparing patients; engaging hospital employees to involve patients; and ensuring leadership support and strategic focus [64]. This model emphasizes the importance of supporting hospital employees and having a strategic PE focus in incorporating PE into hospital activities. The authors of the ECE framework recognize the combined impact of all three pillars in encouraging culture change toward meaningful PE. The findings of this scoping review provide concrete examples of key enablers for PE from leaders, such as active participation in PE committees [37], encouraging policies and procedures that promoted PE [36, 44, 46], communicating PE initiatives and each employee's contributions [46], sharing PE goals [47], showing the benefits of patient partnerships for the hospital and professional practices [50], and being aware of the challenges employees face implementing PE in their work [44, 50]. The findings also provide concrete examples of key enablers for strengthening the capabilities of hospital employees toward PE, such as aligning goals across hospital documents and strategies, aligning processes (e.g., PE in committees, hiring processes, employee performance assessments, and workflow and practices), and employee personal factors (e.g., financial resources, training, tools, physical space, and technologies).

The findings of this scoping review also identify the strategies, resources, and activities employees require to integrate MVV statements into their practice. For example, Slåtten et al. (2021) investigated organizational vision integration, which they defined as "a behavioral manifestation of organizational vision in employee's day-to-day work" [2]. Slåtten et al. found that organizational vision integration mediated the relationship between employee psychological

capital and organizational attractiveness and performance [2]. In another study, Slåtten et al. (2022) found that organizational vision integration was related to employees' organizational commitment [62]. The authors suggested that improving organizational vision integration involves focusing on employees' commitment to their organization by creating an employee-focused organizational culture and having leaders embody MVV statements and empower employees [62]. Previous research has consistently shown that communication that focuses on and encourages employees to integrate hospital goals into their work leads to higher performance and job satisfaction [65]. Having hospital goals concentrating on PE or related concepts (i.e., patient-centered care and patient experience) can inspire, motivate, and empower employees to integrate PE into their work. Having hospital PE goals may also reduce burnout, role strain, and resistance to change, which existed when employees were confused about their hospitals' multiple competing priorities [41, 49, 52, 55, 56]. Furthermore, having employees engage with patients or in the goal development process increases employee motivation and commitment toward PE in their work [37, 38, 41, 50].

This scoping review aligns with and builds upon the ECE framework and studies on organizational vision integration by specifying some strategies, activities, and outcomes for realizing hospital PE goals, such as communicating hospital goals, aligning goals across organizational documents, aligning processes, providing resources and support to hospital employees, and motivating and empowering employees. This scoping review identified numerous ways misalignment impacts employee motivation and commitment to hospital PE goals by analyzing strategies, activities, and outcomes. For example, we found that an organizational imperative to standardize care contradicts and counters efforts to implement person-centered care. The following sections explore the potential effects of misalignment across hospital goals, strategies, and employee values and practices.

## What happens when there is misalignment?

Scoping review findings revealed misalignment between types of goal messages (i.e., patient-centered care versus financial profitability), documents (i.e., patient experience goals in organizational documents vs. care standardization goals in clinical documents), and employee values and practices. Misalignment in the types of goal messages and documents led to employee resistance and confusion, which employees believed was traced to be organizational imperative to standardize care rather than improve patient and family care experiences [41], and a lack of investment in PE infrastructure [55].

These findings are not surprising and have been observed in other studies. For example, Kukkurainen et al. [61] found that while nurses recognized patient-centredness as a core element of their work, implementing person-centered care was constrained by an organizational orientation toward case standardization and cost containment. In this study, the hospital communicated competing priorities (i.e., a commitment to patient-centered care versus promoting a cost-effective culture in ways that may have instead encouraged care standardization) that may have caused tension among nurses and potentially complicated the realization of hospital PE goals.

Balancing the emphasis on different hospital goals is an organizational challenge impacting program, team, and individual behavior. The focus on specific goals must match hospital employees' values, beliefs, and work. Employees who believe that financial goals are more critical than PE goals may feel less motivated toward PE, which may be due to confusion or ambiguity in goal alignment. Alternatively, teams and individuals may be more motivated and empowered to integrate PE into their work if PE goals are communicated and emphasized in organizational communications, documents, and strategies [61]. Employees who believe PE is

a core element of their work may be encouraged when hospitals commit to PE in MVV statements, strategies, and other documents. When hospital employees believe PE is a core element of their work, hospital leadership must augment effective and clear PE goal communication with resources, support, and tools that motivate and empower employees to integrate PE into their work. This scoping review builds on previous frameworks and research by detailing specific organizational strategies to augment effective and clear PE goal communication.

## The complexity of the connection between hospital PE goals and PE practices

The findings of this scoping review suggest that inspiring change and realizing goals require aligning organizational strategies, programs, and team and individual values or goals. However, this is complex and requires disentangling enabling factors, barriers, and outcomes across the organizational, department, individual, and contextual levels. For future work, clarification is needed on whether alignment is required across all organizational elements or just some, including to what degree. Operationalizing the realization factors in hospitals might address misalignment and improve the connection between hospital PE goals and practices.

While this scoping review provided an overview of various factors that impact PE implementation from the perspectives of leaders, hospital employees, and patients, a deeper qualitative analysis is needed to examine these factors in more detail. This is mainly because we found a significant overlap between the realization factors. This suggests that a combination of factors is necessary to understand the relationship between goals and practices. The connection between hospital PE goals and practices is likely intertwined with the teams' and employees' attitudes, values, and routine behaviors. Future research must clarify how different goal messages, documents, and employee beliefs and values affect the connection between hospital PE goals and practices. However, previous frameworks (e.g., the ECE framework) and studies on this topic do not address underlying factors. A new framework is needed that disentangles tacit goals, expectations, and activities. For example, institutional logics categorizes the patterned behaviors of individuals, organizations, and systems [66]. Unlike previous frameworks and studies on this topic, institutional logics provide a language for examining how hospitals may behave in relation to their organizational goals [67] through an analysis of perspectives, artifacts, symbols, and experiences. Specifically, institutional logics can help to examine how macro-level concepts–such as hospital PE goals–can influence employee behavior [68]. Institutional logics thus provides a valuable approach to explore the potential effects of misalignment across hospital PE goals, strategies, and employee values and practices, and also how alignment can support the implementation of PE in practice.

## Limitations of this scoping review

This scoping review has two limitations. First, while we aimed to unpack the connection between hospital PE goals and PE practices, included articles were seldom explicit about this connection. Using deductive analysis with the realization factors framework [28] and inductive analysis to elaborate on specific organizational strategies and outcomes, we could only examine the implicit connection between hospital PE goals and PE practices in some instances. However, there is a need for more primary research that explicitly focuses on understanding and clarifying the connection between hospital PE goals and PE practices, including the role of the realization factors in this connection.

Second, while we sought to stay consistent with the authors' descriptions of organizational goals, there was a lack of consensus regarding conceptualizations of MVV statements [45, 47, 49, 50], strategic priorities [37, 45, 50, 52, 53], values and care principles [38, 53], and a

"community orientation" [48]. We recognized that it might be challenging to conclude the connection between hospital PE goals and PE practices due to diverse conceptualizations of organizational goals. However, incorporating different representations of organizational goals allowed us to create a broader representation of organizational strategies. Future research should focus on examining this relationship with clear and consistent language.

In conclusion, this scoping review explored the connection between hospital PE goals (such as in MVV statements) and PE processes and practices. After an analysis of 27 articles, this scoping review outlined detailed five types of strategies that strengthen the connection between hospital PE goals and practices: communicating organizational goals; aligning documents that convey organizational goals; aligning organizational processes to support PE; providing employees with resources and support; and motivating and empowering employees to integrate PE into their work. However, there is a need for more research to clarify how misalignment between goal messages, documents, and employee beliefs and values affects the connection between hospital PE goals and practices.

## Supporting information

**S1 File.**
(DOCX)

**S1 Checklist. Preferred Reporting Items for Systematic reviews and Meta-Analyses extension for Scoping Reviews (PRISMA-ScR) checklist.**
(DOCX)

## Acknowledgments

We would like to acknowledge Antonia Giannarakos for her support in reviewing the search strategy for this review. Kuluski holds the Dr. Mathias Research Chair in Patient and Family Centred Care. Steele Gray holds a Tier 2 Canada Research Chair in Implementing Digital Health Innovation.

## Author Contributions

**Conceptualization:** Umair Majid, Carolyn Steele Gray, Marianne Saragosa, Pia Kontos, Kerry Kuluski.

**Data curation:** Umair Majid.

**Formal analysis:** Umair Majid, Marianne Saragosa.

**Funding acquisition:** Umair Majid, Kerry Kuluski.

**Investigation:** Umair Majid, Marianne Saragosa, Pia Kontos, Kerry Kuluski.

**Methodology:** Umair Majid, Carolyn Steele Gray, Marianne Saragosa, Pia Kontos, Kerry Kuluski.

**Project administration:** Umair Majid, Carolyn Steele Gray, Pia Kontos, Kerry Kuluski.

**Resources:** Umair Majid, Carolyn Steele Gray, Kerry Kuluski.

**Software:** Umair Majid.

**Supervision:** Umair Majid, Carolyn Steele Gray, Pia Kontos, Kerry Kuluski.

**Validation:** Umair Majid, Marianne Saragosa.

**Visualization:** Umair Majid.

**Writing – original draft:** Umair Majid, Marianne Saragosa.

**Writing – review & editing:** Umair Majid, Carolyn Steele Gray, Marianne Saragosa, Pia Kontos, Kerry Kuluski.

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
