## [Decision Letter · Decision Letter 0]

4 Sep 2023

PONE-D-23-10228Understanding the Connection between Hospital Goals and Patient and Family Engagement: A Scoping ReviewPLOS ONE

Dear Dr. Majid,

Thank you for submitting your manuscript to PLOS ONE. After careful consideration, we feel that it has merit but does not fully meet PLOS ONE’s publication criteria as it currently stands. Therefore, we invite you to submit a revised version of the manuscript that addresses the points raised during the review process.

We look forward to receiving your revised manuscript.

Kind regards,

Fatma Refaat Ahmed, Ph.D.

Academic Editor

PLOS ONE

Journal Requirements:

Reviewers' comments:

Reviewer's Responses to Questions

**Comments to the Author**

1. Is the manuscript technically sound, and do the data support the conclusions?

Reviewer #1: Yes

2. Has the statistical analysis been performed appropriately and rigorously? 

Reviewer #1: N/A

3. Have the authors made all data underlying the findings in their manuscript fully available?

Reviewer #1: Yes

4. Is the manuscript presented in an intelligible fashion and written in standard English?

Reviewer #1: Yes

5. Review Comments to the Author

Reviewer #1: This is a very well written and sophisticated scoping review of an important topic in healthcare today. It demonstrates an excellent understanding of the healthcare organisational problem, scoping review methodology and the use of theoretical and analytical frameworks to aid the review process. The links to additional research in the discussion is particularly noteworthy.

Just some very minor revisions needed.

First sentence of research results is a bit ambiguous. Can this be rephrased please?

The paragraph after Table 2 requires some clarity of writing. “This section” (do the authors mean this scoping review or the original realization factors?). I needed to read this paragraph a couple of times to fully understand the authors intent. Perhaps a sub-heading for this section: Aligning the SR with the original realization factors framework etc..would be helpful.

Barriers / Strategies: I feel that signposting at the start of each realization factor heading, that there was, for example, one major barrier elucidated from the review and five strategies (then numbering each e.g. Strategy 1, strategy 2 etc…) will create additional clarity to the manuscript. As an example, I would therefore suggest bringing forward the paragraph “In summary on page 10…. to become the preamble to the section on Barrier and Strategies on page 9 rather than the conclusion. Same for the motivation and empowerment sections. I like the approach used just make it explicit and upfront for the reader.

Thank you for the opportunity to review this manuscript.

6. PLOS authors have the option to publish the peer review history of their article (what does this mean?). If published, this will include your full peer review and any attached files.

Reviewer #1: No

---

## [Author Response · Author response to Decision Letter 0]

13 Sep 2023

Reviewer #1: This is a very well written and sophisticated scoping review of an important topic in healthcare today. It demonstrates an excellent understanding of the healthcare organisational problem, scoping review methodology and the use of theoretical and analytical frameworks to aid the review process. The links to additional research in the discussion is particularly noteworthy.

Thank you for your positive comments. We are excited that you found this an interesting and important study. 

Just some very minor revisions needed.

First sentence of research results is a bit ambiguous. Can this be rephrased please?

We have revised the initial sentence for clarity. It now reads as “The initial and updated searches combined yielded a total of 8992 abstracts after removing duplicates” (p. 4)

The paragraph after Table 2 requires some clarity of writing. “This section” (do the authors mean this scoping review or the original realization factors?). I needed to read this paragraph a couple of times to fully understand the authors intent. Perhaps a sub-heading for this section: Aligning the SR with the original realization factors framework etc..would be helpful.

Thank you for this comment. We agree that this section needed clarity. We have clarified the writing and hope our intention is clearer in the revised manuscript. (p. 5)

The paragraph now reads as:

“The original realization factors framework lacks clear operationalization because it only offers broad definitions of each factor and a few examples of organizational strategies. In the following sections, based on our scoping review, we detail specific organizational strategies and outcomes under each realization factor to help unpack the connection between hospital goals (MVV statements, strategic plans, and other representations of goals) and PE practices (e.g., aligning goals across organizational documents, elaborating on strategies that align processes with hospital PE goals, and describing ways to provide support and resources to employees to integrate PE into their work).”

Barriers / Strategies: I feel that signposting at the start of each realization factor heading, that there was, for example, one major barrier elucidated from the review and five strategies (then numbering each e.g. Strategy 1, strategy 2 etc…) will create additional clarity to the manuscript. As an example, I would therefore suggest bringing forward the paragraph “In summary on page 10…. to become the preamble to the section on Barrier and Strategies on page 9 rather than the conclusion. Same for the motivation and empowerment sections. I like the approach used just make it explicit and upfront for the reader.

Thank you for this comment. We agree that there is a need for better clarity in the signposting of each subsection of the findings section. We have addressed this by elaborating our signposting by clarifying the specific barriers and strategies discussed in each subsection. We also have a summary table at the beginning of the synthesis section that describe each barrier and strategy (pgs. 7-12).

---

## [Editor Report · Decision Letter 1]

4 Oct 2023

Understanding the Connection between Hospital Goals and Patient and Family Engagement: A Scoping Review

PONE-D-23-10228R1

Dear Dr. Majid,

We’re pleased to inform you that your manuscript has been judged scientifically suitable for publication and will be formally accepted for publication once it meets all outstanding technical requirements.

Kind regards,

Fatma Refaat Ahmed, Ph.D.

Academic Editor

PLOS ONE
---

## [Editor Report · Acceptance letter]

13 Oct 2023

PONE-D-23-10228R1 

Understanding the connection between hospital goals and patient and family engagement: A scoping review 

Dear Dr. Majid:

I'm pleased to inform you that your manuscript has been deemed suitable for publication in PLOS ONE. Congratulations! Your manuscript is now with our production department. 

Kind regards, 

on behalf of

Dr. Fatma Refaat Ahmed 

Academic Editor

PLOS ONE